# Efficient Algorithms for Lipschitz Bandits

## Abstract

Lipschitz bandits is a fundamental framework used to model sequential decision-making problems with large, structured action spaces. This framework has been applied in various areas. Previous algorithms, such as the Zooming algorithm, achieve near-optimal regret with $O(T^2)$ time complexity and $O(T)$ arms stored in memory, where $T$ denotes the size of the time horizons. However, in practical scenarios, learners may face limitations regarding the storage of a large number of arms in memory. In this paper, we explore the bounded memory stochastic Lipschitz bandits problem, where the algorithm is limited to storing only a limited number of arms at any given time horizon. We propose algorithms that achieve near-optimal regret with $O(T)$ time complexity and $O(1)$ arms stored, both of which are almost optimal and state-of-the-art. Moreover, our numerical results demonstrate the efficiency of these algorithms.

## 1  Introduction

Multi-armed Bandits (MAB) is a powerful framework used to balance the exploration-exploitation trade-off in online decision-making problems. Within this framework, a learner sequentially selects arms (actions, decisions, or items) and learns from the associated feedback, aiming to maximize the expected total reward within finite time horizons. Some well-known algorithms, such as UCB1 and Exp3, have achieved near-optimal regret by storing records of all arms in memory. In many bandit problems, algorithms can access information about the similarity between arms, suggesting that arms with similar characteristics often yield similar expected rewards. The Lipschitz bandits framework is a prominent variant that addresses decision-making in large, structured action spaces, where the expected reward of the arms follows a Lipschitz function. For instance, in recommendation systems, the arms correspond to items represented by feature vectors. Items with similar feature vectors are likely to result in similar outcomes or conversions.

Recently, a series of works in the field of online learning have been dedicated to managing scenarios with large action spaces while maintaining sub-linear memory usage. This direction is driven by the need to effectively tackle extensive real-world applications such as recommendation systems, search ranking, and crowdsourcing. In these applications, arms correspond to items, solutions, or models, which leads to significant memory demands. For instance, in recommendation systems, the learner faces the challenge of choosing from millions of items, like music and movies, to present to users, especially in scenarios characterized by limited space or an infinite number of arms. Therefore, the development of memory-efficient algorithms has become crucial for these applications. In recent years, substantial efforts have been made to address the challenge of bandits with limited memory (Assadi & Wang, 2020; Jin et al., 2021; Maiti et al., 2020; Agarwal et al., 2022; Assadi & Wang, 2022; Wang, 2023; Assadi & Wang, 2023a). However, previous research has mainly focused on unstructured action spaces, often overlooking the fact that in these applications, arms with similar characteristics tend to yield similar expected rewards.

Table 1: Comparison with State-of-the-art Lipschitz Bandits Algorithms

| Algorithm | Regret | Time complexity | Space complexity |
|---|---|---|---|
| Zooming(Kleinberg et al., 2019) | $\widetilde{O}\left(T^{\frac{d_z+1}{d_z+2}}\right)$ | $O\left(T^2\right)$ | $O(T)$ |
| HOO(Bubeck et al., 2011a) | $\widetilde{O}\left(T^{\frac{d_z+1}{d_z+2}}\right)$ | $O\left(T\log T\right)$ | $O(T)$ |
| MBAD(Ours) | $\widetilde{O}\left(T^{\frac{d_z+1}{d_z+2}}\right)$ | $O\left(T\right)$ | $O\left(1\right)$ |

One general approach to solving Lipschitz bandits is through discretizing the structured action space. Algorithms based on uniform discretization have been shown to achieve optimal worst-case regret up to a logarithmic factor (Kleinberg, 2004). Another strategy, adaptive discretization, progressively 'zooms in' on more promising regions of the action space, yielding near-optimal problem-dependent regret (Kleinberg et al., 2019). However, existing algorithms like the Zooming algorithm necessitate $O(T)$ stored arms in memory and $O(T^2)$ time complexity for stochastic Lipschitz bandits (Kleinberg et al., 2019; Feng et al., 2022), which may be impractical for many real-world applications. In this paper, we consider a typical scenario where the learner operates within the stochastic bandits framework over a Lipschitz action space while facing constraints on the number of arms that can be stored in memory.

The limited memory constraint and large structured action space present several challenges, necessitating a nuanced approach to effectively balance exploration and exploitation under uncertainty. One key challenge is the propensity to over-exploit suboptimal arms retained in memory, leading to high regret. Conversely, reading new arms into memory risks discarding potentially valuable arms. In scenarios with infinite actions, the vast search space requires numerous samples to ensure adequate exploration. The structured nature of the action space demands that algorithms focus on zooming in on more promising regions, but space constraints limit the learner's capacity to acquire comprehensive knowledge about the metric space. Traditional full-memory algorithms start by dividing the action space into many small subcubes, a process known as discretization. Each cube is treated as an arm, and in each round, the algorithm updates the average estimate of the selected cube's reward based on feedback. It then compares this estimate against all other cubes in the storage space through various computational methods.

## 1.1 Our Contributions

Our primary insight revolves around two key aspects: metric embedding and pairwise comparisons. Metric embedding involves mapping elements from one metric space to another while preserving distance relationships as closely as possible. Our algorithm effectively maps the metric space to a tree, where each node represents a cube. Traversing this tree is analogous to navigating the entire metric space. Pairwise comparisons of arms reduce memory complexity. Instead of constantly covering the entire space, our approach considers all subcubes as a stream. From this stream, we continuously select cubes for pairwise comparisons, gradually converging to the optimal region.

Based on this insight, we introduce two algorithms: the Memory Bounded Uniform Discretization (MBUD) algorithm and the Memory Bounded Adaptive Discretization (MBAD) algorithm. The MBUD algorithm employs a uniform discretization strategy combined with an Explore-First approach. In this method, all cubes are of the same size. The algorithm prioritizes selecting a near-optimal arm following an exploration phase and allocates the remaining rounds to exploitation, achieving near-optimal worst-case regret. The exploration phase consists of "cross exploration phases" and the "summarize phase". During the cross exploration phases, exploration is confined to a subset of cubes to gather information about the optimal arm while minimizing regret. The summarize phase explores all cubes to pinpoint the optimal arm's location.

The MBAD algorithm utilizes an adaptive discretization strategy, incorporating a round-robin playing approach. This allows for subcubes within subcubes, organizing the entire action space into a tree structure. The algorithm selectively focuses on more promising regions of the action space, thereby attaining near-optimal instance-dependent regret. Each node in this structure represents a subcube, with parent and child nodes corresponding to subcubes and their subdivisions, respectively. Traversal involves transitioning from a node to its child and navigating through a parent node's children to the next subcube. Pruning prevents over-zooming through two conditions: discarding inferior cubes with high confidence and establishing a lower bound on cube edge length, which decreases as exploration progresses. These conditions ensure efficient exploration without over-zooming.

Overall, our contribution lies in pioneering memory-efficient algorithms for large structured action spaces, particularly within Lipschitz metric spaces. We introduce the MBUD and MBAD algorithms, which achieve near-optimal regret while requiring storage for only the best-estimate arm for exploitation and one additional arm for exploration. This means only two arms need to be stored in memory, regardless of the problem's scale. Furthermore, each algorithm exhibits $O(T)$ time complexity, indicating that their execution time scales linearly with the number of rounds.

## 1.2 Related Work

**Lipschitz bandits.** Multi-armed bandits is one of the most classical frameworks to model the trade-off between exploration and exploitation in online decision problems. The Lipschitz bandits framework considers the large, structured action space in which the algorithm has information on similarities between arms. The model was first introduced by Agrawal (1995) with interval $[0, 1]$. The near-optimal upper and lower bounds for the worst case were provided in Kleinberg (2004) via the uniform discretization strategy. Subsequent work (Kleinberg et al., 2019) proposed the zooming algorithm, achieving near-optimal instance-dependent regret for the problem and studying the extension for the general metric action space. Several other works have established regret bounds for the stochastic reward feedback setting (Bubeck et al., 2011a; Magureanu et al., 2014; Lazaric et al., 2014). Other works have also extended the results to the adversarial version (Podimata & Slivkins, 2021; Kang et al., 2023), contextual setting (Slivkins, 2014; Krishnamurthy et al., 2019; Lee et al., 2022), ranked setting (Slivkins et al., 2013), contract design (Ho et al., 2014), federated X-armed bandit (Li et al., 2024a,b), and other settings (Bubeck et al., 2011b; Lu et al., 2019; Wang et al., 2020; Grant & Leslie, 2020; Feng et al., 2022; Xue et al., 2024).

**Memory-efficient learning.** Another line relevant to this paper is online learning with memory constraints. Liau et al. (2018) considered stochastic bandits with constant arm memory and proposed an algorithm achieving an $O(\log 1/\Delta)$ factor of optimal instance-dependent regret, where $\Delta$ is the gap between the best arm and the second-best arm. Chaudhuri & Kalyanakrishnan (2020) studied stochastic bandits with $M$ stored arms and showed there is an algorithm with regret $\tilde{O}(KM + (K^{3/2}\sqrt{T})/M)$. Subsequent work (Agarwal et al., 2022) provided an algorithm achieving regret $O(\sqrt{KT \log T \log \log T})$. In addition to the bandits problem, there are also many works about other online learning problems. Srinivas et al. (2022); Peng & Zhang (2022) showed the trade-off between regret and memory for the expert problem. More pure exploration models with memory constraints were considered in Assadi & Wang (2020), including the coin tossing problem, noisy comparisons problem, and Top-$K$ arms identification. Previous works on bandits with limited memory have not considered structured action spaces and could not deal with infinite actions. There are some other works on memory-efficient online learning (Peng & Rubinstein, 2023; Assadi & Wang, 2023b). Beyond the online learning setting, the memory-efficient learning problem was solved in different situations, including statistical learning (Steinhardt et al., 2016; Garg et al., 2017; Raz, 2017; Garg et al., 2019; Sharan et al., 2019; Lyu et al., 2023), convex optimization (Marsden et al., 2022; Blanchard et al., 2023a,b; Chen & Peng, 2023), estimation problems (Acharya et al., 2019; Diakonikolas et al., 2022; Berg et al., 2022), parity learning (Raz, 2019; Kol et al., 2017), and other learning problems (Hopkins et al., 2021; Brown et al., 2022; Chen et al., 2022).

## 2 Problem Setup and Preliminaries

**Notations.** In this paper, we use bold fonts to represent vectors and matrices. For a positive integer $T$, we use $[T]$ to denote the set $\{1, 2, \ldots, T\}$. For a set $\mathcal{X}$, we use $|\mathcal{X}|$ to denote its cardinality. For a random variable $Z$, we use $\mathbb{E}[Z]$ to denote its expectation. For an event $\mathcal{E}$, we use $\mathbb{P}[\mathcal{E}]$ to denote its probability.

## 2.1 Problem Setup

We formally define the Lipschitz bandits problem below. Given $T$ rounds, dimension $d$, and arm space $\mathcal{X} = [0, 1]^d$, each arm $x \in \mathcal{X}$ is associated with an unknown reward distribution $\mathcal{D}_x$. In each round $t \in [T]$, the algorithm selects an arm $x_t \in \mathcal{X}$ and obtains a scalar-valued reward feedback $r_t \in [0, 1]$, which is a sample from the reward distribution $\mathcal{D}_{x_t}$. The expected reward $\mu(\cdot)$ of the reward distribution satisfy the Lipschitz condition: $|\mu(x) - \mu(y)| \leq L \cdot |x - y| \quad \forall x, y \in \mathcal{X}$. And we call $L$ the Lipschitz constant. Then a problem instance is specified by the known number of time horizons $T$, known Lipschitz constant $L$, and unknown mean reward $\mu(\cdot)$. For the purposes of simplification in our proofs, we assume $L = 1$. The algorithm aims to maximize the expected total reward $\mathbb{E}[\sum_{t \in [T]} r_t]$.

We use regret to measure the performance of the algorithm compared with the expected total reward of the best-fixed arm in action space $\mathcal{X}$: $\mathbb{R}_{\mathcal{X}}(T) = T \cdot \sup_{x \in \mathcal{X}} \mu(x) - \mathbb{E}\left[\sum_{t \in [T]} r_t\right]$.

Then we present the memory model employed in the paper. The algorithm operates by selecting arms from the memory and pulling them. When the memory reaches the capacity and the algorithm attempts to choose a new arm, it becomes necessary to discard at least one arm from the memory. Consequently, any statistical information associated with the discarded arm, including its index, mean reward, and number of pulls, is forgotten and will not be retained thereafter. We measure the space complexity of the algorithm by the hard constraint for the number of arms stored in the memory. This constraint aligns with the assumption of having oracle access to the input arm, as commonly defined in streaming problems.

## 2.2 Covering Dimension and Zooming Dimension

Then we provide some technical tools that are used in this paper and introduce the covering dimension and zooming dimension for one action space $\mathcal{X}$. We use the definitions in (Slivkins, 2019) and provide them below. Notice that the Lipschitz bandits problem is defined in an infinite-action space. We select a fixed, finite discretization actions space $\mathcal{S} \subset \mathcal{X}$. Let $\{\mathcal{X}_1, \ldots, \mathcal{X}_N\}[\mathcal{X}_i \subset \mathcal{X}]$ be an cover of the action space $\mathcal{X}$. Let $\epsilon$ denote the maximum diameter of $\mathcal{X}_i$ for all $i \in [N]$. Then the arm set $\mathcal{S} = \{x_i | x_i \in \mathcal{X}_i, i \in [N]\}$ is an $\epsilon$-mesh. The covering dimension $d$ of the action space $\mathcal{X}$ is defined as $d = \inf_{\alpha \geq 0} \{|\mathcal{S}| \leq \epsilon^{-\alpha}, \forall \epsilon > 0\}$. Let $\mu^*_{\mathcal{X}} := \sup_{x \in \mathcal{X}} \mu(x)$ denote the expected per-round reward of the optimal arm in space $\mathcal{X}$ and $\Delta(x) := \mu^*_{\mathcal{X}} - \mu(x)$ denote the gap between arm $x$ and the optimal arm. Define $\mathcal{Y}_j = \{x \in \mathcal{X} : 2^{-j} \leq \Delta(x) < 2^{1-j}, j \in \mathbb{N}\}$, then set $\mathcal{Y}_j$ contains all arms whose gap is between $2^{-j}$ and $2^{1-j}$. Consider the $\epsilon$-mesh $\mathcal{S}_j$ for space $\mathcal{Y}_j$. Then the zooming dimension $d_z$ for the action space $\mathcal{X}$ is $d_z = \inf_{\beta \geq 0} \{|\mathcal{S}_j| \leq \epsilon^{\beta}, \epsilon = 2^{-j}, \forall j \in \mathbb{N}\}$.

Covering dimension is a property of the action space while the zooming dimension is a property of the instance. Notice that we always have $d_z \leq d$. This is because the covering dimension considers the $\epsilon$-mesh of the entire action space $\mathcal{X}$, whereas the zooming dimension focuses only on the set $\mathcal{Y}_j$. The covering dimension is closely related to other notions of dimensionality in a metric space, such as the Hausdorff dimension, capacity dimension, and box-counting dimension, all of which characterize the covering properties in fractal geometry. Similarly, the zooming dimension is another measure used to evaluate the structure of a metric space. Both of these dimensions are widely utilized in the field of Lipschitz bandits. For further details and alternative formulations regarding the covering dimension and zooming dimension, refer to (Kleinberg et al., 2019).

# 3 Warm Up: Uniform Discretization Algorithm

This section provides the intuition, specification, and theoretical analysis of the Memory Bounded Uniform Discretization (MBUD) algorithm (shown in Algorithm 1) for the stochastic Lipschitz bandits problem.

**Algorithm overview.** To facilitate our discussion, we begin by outlining the core idea behind the algorithm. This algorithm employs a uniform discretization strategy and adopts an Explore-First methodology, which endeavors to identify a near-optimal arm following the exploration phase and dedicates the remaining rounds to exploitation. Throughout the exploration stage, the algorithm allocates two units of memory space: one for storing the best-estimated arm and another for temporarily holding a newly read arm. Note that the best-estimated arm serves a dual purpose: it is not only crucial for the exploitation phase but also enables the swift identification of sub-optimal arms.

The exploration phase in Algorithm 1 is divided into $\lceil \log \log T \rceil$ phases, further structured into two main segments: the 'cross exploration phases' and the 'summarize phase'. During the initial $\lceil \log \log T \rceil - 1$ phases, the algorithm iterates over the arms within the discretized action space to minimize regret. Exploration is limited to a subset of cubes at a time, allowing the algorithm to gather information about the optimal arm while minimizing regret. In the final phase, termed the 'summarize phase', the algorithm revisits all arms within the uniform discretization space. Overall, each arm is read into memory twice to ensure thorough evaluation. Furthermore, we implement a budgeting strategy for each phase, wherein the total number of pulls across all arms is constrained by a predefined budget. The goal is to select the optimal arm with high probability after accumulating sufficient information during the previous phases. This structured approach balances exploration and

---

**Algorithm 1** Memory Bounded Uniform Discretization (MBUD)

---

**Input:** arm space $\mathcal{X} = [0,1]^d$, time horizon $T$, parameter $c$.

1: $\boldsymbol{y} \leftarrow \boldsymbol{0}, \bar{r}_y \leftarrow 0, n_y \leftarrow 0, B_{-1} \leftarrow 1, \epsilon = \left(\frac{\log T}{T}\right)^{1/(d+2)}, \phi \leftarrow \lceil \log\log T \rceil - 1.$
2: **for** $p = 0, \cdots, \phi - 1$ **do**
3:     $B_p \leftarrow \sqrt{T B_{p-1}}.$
4:     **for** $q = 1, \cdots, \lfloor \phi\epsilon^{-d} \rfloor$ **do**
5:         Generate a new cube $C \leftarrow$ CROSSCUBE$(\phi, \epsilon, p, q)$, and select a arm $\boldsymbol{x}$ from $C$.
6:         $(\boldsymbol{y}, \bar{r}_y, n_y) \leftarrow$ COMPARE$(c, \boldsymbol{x}, \boldsymbol{y}, \bar{r}_y, n_y, \epsilon B_p).$
7:     **end for**
8: **end for**
9: **for** $q = 1, \cdots, \lfloor \epsilon^{-d} \rfloor$ **do**
10:     Generate a new cube $C \leftarrow$ GENERATECUBE$(\epsilon, q)$, and select a arm $\boldsymbol{x}$ from $C$.
11:     $(\boldsymbol{y}, \bar{r}_y, n_y) \leftarrow$ COMPARE$(c, \boldsymbol{x}, \boldsymbol{y}, \bar{r}_y, n_y, \epsilon B_{\phi-1}).$
12: **end for**
13: Play arm $y$ until the end of the game.

---

---

**Algorithm 2** CROSSCUBE

---

**Input:** number of phases $\phi$, edge-length $\epsilon$, parameters $q$.

1: $\kappa_1 \leftarrow \max_{k\in\mathbb{N}}\{k^d \leq \phi\}, \kappa_2 \leftarrow \max_{k\in\mathbb{N}}\{k^d \leq \lfloor \phi\epsilon^{-d} \rfloor\}.$
2: Let node $\leftarrow \epsilon\mathcal{G}_d(p, \kappa_1) + \frac{\epsilon\phi}{\sqrt{d}}\mathcal{G}_d(q, \kappa_2)$, then the cube could be determined by node and $\epsilon$.

---

exploitation under memory constraints, aiming to quickly identify the optimal arm while minimizing the sampling of suboptimal arms. The specifics of this approach will be detailed subsequently.

**Exploration strategies.** For the cross exploration phases, the gap between neighboring arms is $\epsilon\phi$ ($\phi$ defined in Algorithm 1). There are $O(\epsilon^{-d})$ cubes (arms) in the discretization action set, which is an $\epsilon$-mesh of $\mathcal{X}$. Each cross exploration phase will only explore $O\left(\frac{1}{\log\log T}\right)$ of them. We generate a new cube by using the function $\mathcal{G}_d(a,b), a, b \in \mathbb{N}$ which converts the integer $a$ to a $d$-dimension vector. And the $i$-th entry of the vector is the $i$-th right-most digit in base $b$. To aid understanding, we offer several examples: $\mathcal{G}_3(3,2) = (0,1,1)$, $\mathcal{G}_3(1208, 26) = (1, 20, 12)$, and $\mathcal{G}_2(1208, 26) = (20, 12)$. The function could be done by a succession of Euclidean divisions by $b$. For the summarize phase, the gap is $\epsilon$ and all cubes in the discretization set are explored.

The CROSSCUBE function generates cubes for the cross exploration phases by calculating parameters based on the number of phases and the edge-length of the cubes. Specifically, CROSSCUBE generates a new cube using a combination of two geometric sequences. It first calculates the parameters $\kappa_1$ and $\kappa_2$ as the maximum integers such that $k^d \leq \phi$ and $k^d \leq \lfloor \phi\epsilon^{-d} \rfloor$, respectively. The function then determines the cube's position using these parameters and the edge-length $\epsilon$. The cube is defined by a node position generated by $\epsilon\mathcal{G}_d(p, \kappa_1)$ and $\frac{\epsilon\phi}{\sqrt{d}}\mathcal{G}_d(q, \kappa_2)$, where $\mathcal{G}_d$ is a geometric sequence generator that converts an integer to a $d$-dimensional vector. The GENERATECUBE function is similar to CROSSCUBE but is used during the summarize phase to generate cubes without considering the phases. It calculates the parameter $\kappa$ as the maximum integer such that $k^d \leq \lfloor \epsilon^{-d} \rfloor$. The cube is then determined by the edge-length $\epsilon$ and a node position generated by $\epsilon\mathcal{G}_d(q, \kappa)$.

**Compare strategy.** Then we introduce the compare strategy, which is also useful for the MBAD algorithm described in the following section. The algorithm always selects the arm with the fewest pulls in the memory. After sufficient samples, it will eliminate one sub-optimal arm based on its upper confidence bound and then generate a new arm (i.e., read a new arm into the memory). Notice that the algorithm may prioritize two sub-optimal arms with a small gap. Therefore, there is a cap on the number of pulls each phase for any arm. It helps the algorithm in striking a balance between exploration (read a new arm) and exploitation (play arms in memory).

The algorithm maintains three statistics for one arm in memory: the index $x$, the mean reward estimator $\bar{r}_x$, and the number of pulls $n_x$. The constant $c$ is an exploration and exploitation balancing parameter. In the exploration part, there are $\lceil \log\log T \rceil$ phases. Let $B_p$ be the budget of samples for the $p$-th phase. We use $y$ and $x$ to denote the best-estimated arm and the new arm in the algorithm,

---

**Algorithm 3** GENERATECUBE

---

**Input:** edge-length $\epsilon$, parameters $q$.

  1: $\kappa \leftarrow \max_{k \in \mathbb{N}}\{k^d \leq \lfloor \epsilon^{-d} \rfloor\}$.
  2: Let $\texttt{node} \leftarrow \epsilon \mathcal{G}_d(q, \kappa)$, then the cube could be determined by $\texttt{node}$ and $\epsilon$.

---

---

**Algorithm 4** COMPARE

---

**Input:** constant $c$, arm $x$ and $y$, $\bar{r}_y$, $n_y$, $b$.

  1: $\bar{r}_x \leftarrow 0$, $n_x \leftarrow 0$.
  2: **while** $n_x \leq b$ or $n_y \leq b$ **do**
  3:     Pull the least played arm between $x$ and $y$. If there is no single least played arm, select a random arm.
  4:     Update $\bar{r}_x, n_x, \bar{r}_y, n_y$.
  5:     **if** $\min\{\bar{r}_x + \sqrt{(c \log T)/n_x}, 1\} < \max\{\bar{r}_y - \sqrt{(c \log T)/n_y}, 0\}$ **then**
  6:       Break and return $(y, \bar{r}_y, n_y)$.
  7:     **else if** $\max\{\bar{r}_x - \sqrt{(c \log T)/n_x}, 0\} > \max\{\bar{r}_y - \sqrt{(c \log T)/n_y}, 0\}$ **then**
  8:       Break and return $(x, \bar{r}_x, n_x)$.
  9:     **end if**
10: **end while**
11: Return $(y, \bar{r}_y, n_y)$.

---

respectively. If the upper confidence bound (UCB) of arm $x$ is less than the lower confidence bound (LCB) of arm $y$, then $x$ is suboptimal with high probability. If the LCB of $y$ is less than the LCB of arm $x$, then $x$ is not too bad with high probability. For the remaining cases, we could choose either $x$ or $y$, and we choose arm $y$ at the end of the algorithm.

**Flowchart.** In Appendix A.1, we include a flowchart that illustrates the operation of the algorithm.

**Theoretical result.** The computational workload of the MBUD algorithm is characterized by a constant per-round operation, leading to a total time complexity of $O(T)$, where $T$ represents the number of rounds. Regarding space complexity, the MBUD algorithm necessitates the storage of merely two arms in memory at any given time. Additionally, the space requirements for the GENERATECUBE and CROSSCUBE subroutines are minimal, each consuming $O(1)$ units of space in terms of arm storage. Consequently, the overall space complexity of the algorithm is $O(1)$.

We provide the theoretical result below and provide the details of the theoretical analysis in Appendix B. The result recovers the worst case regret in previous work and recovers the lower bound up to a logarithmic factor (Kleinberg, 2004).

**Theorem 1.** *For the stochastic Lipschitz bandits problem with metric $(\mathcal{X}, \mathcal{D})$ and time horizon $T$, where $\mathcal{X} = [0,1]^d$ and $\mathcal{D}$ is a known metric function. Algorithm 1 uses $O(1)$ stored arms and achieves regret*

$$\mathbb{R}_{\mathcal{X}}(T) \leq \tilde{O}(T^{\frac{d+1}{d+2}}),$$

*where $d$ is the covering dimension of space $\mathcal{X}$.*

The theoretical analysis is mainly based on the 'clean event', which holds that the observed mean average is a good estimator for the expectation with high probability. At a high level, the analysis shows that the deviation between the mean estimator of the best-estimated arm $y$ and the optimal expected reward $\mu_{\mathcal{X}}^*$ is small enough when $p \geq 1$. Then the sub-optimal arms could be discarded quickly, which helps us to bound the incurred regret of sub-optimal arms and the exploitation phase. We bound the expected regret during all time horizons by considering the discretization error, the incurred regret of all sub-optimal arms during the exploration, and the sub-optimality of the selected arm before the exploitation together.

## 4 Adaptive Discretization Algorithm

This section provides the main idea, specification, and theoretical analysis of the Memory Bounded Adaptive Discretization (MBAD) algorithm (shown in Algorithm 5).

---

**Algorithm 5** Memory Bounded Adaptive Discretization (MBAD)

---

**Input:** time horizon $T$, constant $c$.

1: $y \leftarrow 0, \bar{r}_y \leftarrow 0, n_y \leftarrow 0, B_1 \leftarrow \sqrt{T}$.
2: **for** $p = 1, 2, \ldots$ **do**
3:      $x \leftarrow 0, \bar{r}_x \leftarrow 0, n_x \leftarrow 0, b_p \leftarrow B_p \cdot \left(\frac{\log T}{T}\right)^{1/(d+2)}$.
4:      ADAPTIVECUBE(4, 1).
5:      $B_{p+1} \leftarrow B_p \log T$.
6: **end for**

---

---

**Algorithm 6** ADAPTIVECUBE

---

**Input:** parameters $m, q$, edge-length $\epsilon = 2^{-m}$.

1: $\kappa \leftarrow \max_{k \in \mathbb{N}}\{k^d \leq \lfloor \epsilon^{-d} \rfloor\}$.
2: $\texttt{node} \leftarrow \epsilon \mathcal{G}_d(q, \kappa)$, then the cube $C$ could be determined by $\texttt{node}$ and $\epsilon$.
3: Select a arm $\boldsymbol{x}$ from $C$.
4: **if** $q + 1 \leq 2^m$ **and** the output of COMPARE$(c, \boldsymbol{x}, \boldsymbol{y}, \bar{r}_y, n_y, 20\epsilon^{-2})$ is arm $\boldsymbol{y}$ **then**
5:      check the next cube with parameters $m$ and $q + 1$.
6: **else if** $20\epsilon^{-2} \leq b_p$ **then**
7:      $(\boldsymbol{y}, \bar{r}_y, n_y) \leftarrow$ COMPARE$(c, \boldsymbol{x}, \boldsymbol{y}, \bar{r}_y, n_y, 20\epsilon^{-2})$.
8:      Equally partition the cube $C$ into $2^d$ subcubes and check the first subcube.
9: **end if**

---

**Algorithm overview.** We begin with some intuitions. The MBUD algorithm achieves near-optimal regret in the worst case but fails to leverage the beneficial structure of 'nice' problem instances. To address this, we present the MBAD algorithm, which is based on adaptive discretization, and establish a near-optimal instance-dependent upper bound. The idea behind adaptive discretization is straightforward: the algorithm should focus more on promising regions. For instance, the zooming algorithm approximates the expected rewards over the action space and explores more in regions with a high probability of yielding high rewards. However, due to memory constraints, the algorithm cannot obtain a comprehensive picture of the action space over time. To overcome this obstacle, the MBAD algorithm employs a "round robin" strategy, storing the best-estimated arm as the next read arm in memory. Unlike the MBUD method, which chooses predetermined steps, the MBAD algorithm selects the next read arm based on the confidence radius of the arms in memory. Consequently, steps are smaller and probes (newly picked arms) are more numerous in promising regions.

**Exploration strategies.** The ADAPTIVECUBE subroutine is the cornerstone of the MBAD algorithm, functioning as a recursive mechanism to navigate and leverage a cubic region within the decision space. This procedure dynamically adjusts the exploration granularity based on observed rewards and predetermined sampling constraints. Initially, the algorithm selects a cube $C$ for exploration. If this cube is deemed sub-optimal compared to the optimal estimated arm stored in memory (denoted as arm $\boldsymbol{y}$), the algorithm discards this cube in favor of exploring a subsequent cube, following the generation rules outlined in the GENERATECUBE subroutine described in the MBUD algorithm (Section 3). Conversely, if the cube shows promise, the algorithm proceeds to explore within it, subdividing it into smaller subcubes for more detailed exploration. Each exploration phase is governed by a specific sample budget, which regulates the granularity of exploration to prevent excessive sampling of sub-optimal arms in the early stages. This adaptive exploration process continues until the entire action space has been thoroughly explored. The decision-making process is inherently dynamic, constantly evolving based on past actions to enhance the efficiency of future exploration and exploitation efforts.

To prevent the MBAD algorithm from "over-zooming", we implement two stop conditions. The first condition discards the current cube in favor of a new one once we are highly confident that the current cube is inferior to the best cube we've explored (see lines 4-5 of Algorithm 6). The second condition sets a lower bound on the edge length of the cube to be explored in each round, which gradually decreases as exploration progresses (see line 6 of Algorithm 6). These conditions together ensure the algorithm avoids over-zooming. In the initial learning phase, our knowledge of the optimal cube is limited, making it challenging to effectively distinguish suboptimal cubes using only the first stop

condition. However, the second condition, with a larger initial lower bound on cube edge length, prevents over-zooming. As the learning process advances, the algorithm can more reliably eliminate suboptimal cubes, thus avoiding over-zooming on them.

**Flowchart and algorithm description.** Due to page limitations, Appendix A.2 contains a flowchart illustrating the operation of the algorithm along with its description.

**Theoretical result.** Analyzing the space complexity of the MBAD algorithm and its ADAPTIVE-CUBE subroutine requires careful consideration due to the subroutine's recursive nature. Specifically, the conditional logic that triggers further recursion or partitioning into $2^d$ subcubes adds layers of complexity. Within the ADAPTIVECUBE subroutine, each recursive invocation contributes to the call stack, with space consumption directly proportional to the recursion depth. The space required to sustain the state of each cube, alongside the recursive call stack within ADAPTIVECUBE, implies a complexity that scales linearly with recursion depth, complemented by constant overheads for variables preserved at each recursion level. Nonetheless, the algorithm's design allows for the direct computation of all parent and neighboring cube information from the current cube's coordinates and edge length, obviating the need for multiple cube storage in memory. Consequently, only a single cube needs to be maintained at any time during the ADAPTIVECUBE process, affirming a space complexity of $O(1)$ for the MBAD algorithm. This space complexity analysis directly informs the algorithm's time complexity. Similar to the MBUD algorithm, the overall time complexity of the MBAD algorithm remains linear with respect to the total number of rounds.

As a by-product of the MBAD algorithm, we introduce a simpler, more practical algorithm for scenarios where $d_z \leq 1$. Detailed descriptions and theoretical analyses of this algorithm can be found in Appendix D. We provide the theoretical result below and elaborate on the details of the theoretical analysis in Appendix C. The result establishes the optimal instance-dependent upper bound, up to a logarithmic factor, for the stochastic Lipschitz bandits problem. Previous works (Slivkins, 2014; Kleinberg et al., 2019) have already established related lower bounds, indicating that our work achieves near-optimal regret. While there are other forms of results, such as those presented in work (Magureanu et al., 2014), we believe that adopting one form is sufficient to demonstrate the near-optimal performance of our algorithm.

**Theorem 2.** *For Lipschitz bandits with time horizon $T$ and Lipschitz constant $L$, Algorithm 5 with $c \geq 5$ achieves regret*

$$\mathbb{R}_{\mathcal{X}}(T) \leq \tilde{O}(T^{\frac{d_z+1}{d_z+2}}),$$

*using $O(1)$ stored arms, where $d_z$ is the zooming dimension of space $\mathcal{X}$.*

We also mainly consider the clean event. The algorithm plays in a 'round-robin' manner. There are at most $O(\log T)$ phases because of the delicate design of the budget for each phase. For each phase, we show that the deviation between the mean reward of the best-estimated arm and optimal expected per-round reward $\mu_{\mathcal{X}}^*$ is small. Then the algorithm could approximately adjust the sub-optimality of arms and set more probes in more promising regions. Then we prove that the incurred regret could be bounded by $O(T^{\frac{z+1}{z+2}}(\log T)^{\frac{2}{z+2}})$ by bounding the pulls of bad arms according to the definition of zooming dimension.

## 5 Numerical Evaluations

In this section, we show the efficiency of our algorithms through a series of numerical simulations. The baseline consists of three algorithms: the uniform discretization with UCB1 algorithm (UD) and the zooming algorithm. For the uniform discretization, we pick a fixed $\epsilon$-mesh of the action space and run the UCB1 algorithm only considering the finite uniform discretization action space. The UCB1 algorithm is a popular algorithm for achieving near-optimal regret with finite action space. Kleinberg (2004) prove that the uniform achieves optimal worst-case regret up to logarithm factors. The zooming algorithm (Kleinberg et al., 2019) is an implementation of the adaptive discretization strategy, which deploys more probes in regions deemed more 'promising'. Theoretical analysis shows that the zooming algorithm both achieves optimal worst-case regret and instance-dependent regret up to logarithm factors.

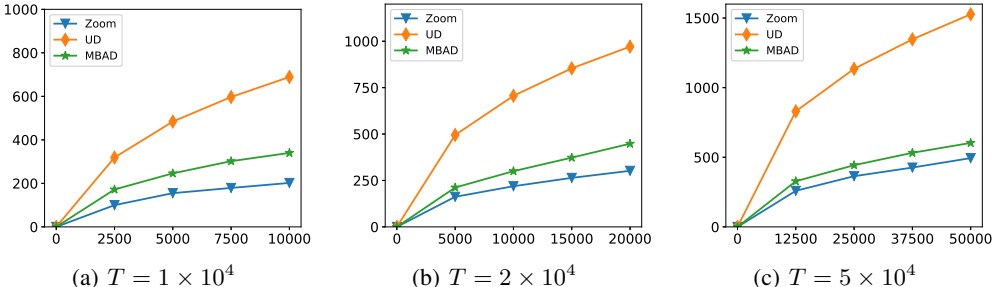



(a) $T = 1 \times 10^4$      (b) $T = 2 \times 10^4$      (c) $T = 5 \times 10^4$

Figure 1: The results obtained with different time horizons.



We set $\mathcal{X} = [0, 1]$ and choose the reward function $f(x) = 0.5 - |x - 0.5|$. In each round $t$, the algorithm plays one arm $x_t$ and receives a stochastic reward $y$ satisfying

$$y = \begin{cases} f(x) + \xi, & 0 \leq f(x) + \xi \leq 1 \\ 1, & f(x) + \xi > 1 \\ 0, & f(x) + \xi < 0 \end{cases}.$$

Specifically, $\xi \sim \mathcal{N}(0, 0.1^2)$ is the Gaussian noise. The MBAD algorithm are only allowed to store two arms in memory, while there is no memory constraint for the UD+UCB1 algorithm and the zooming algorithm. All results are averages over 50 runs. Figure 1 displays the results obtained across varying time horizons, where the horizontal axis denotes the time horizon and the vertical axis measures regret. From the figure, we have that MBAD algorithm significantly outperforms the UD strategy. Additional numerical results are detailed in Appendix E.

## 6  Conclusion and Discussion

We consider the Lipschitz bandits with limited memory problem. We introduce two novel algorithms: the Memory Bounded Uniform Discretization (MBUD) algorithm and the Memory Bounded Adaptive Discretization (MBAD) algorithm, which are predicated on the principles of uniform and adaptive discretization, respectively. Theoretical analyses reveal that the MBAD algorithm achieves near-optimal performance with $O(1)$ stored arms and $O(T)$ time complexity, highlighting its efficiency and practical applicability. Moreover, numerical results show the efficiency of our algorithms.

The Lipschitz bandit problem in higher dimensions is often perceived as a 'needle in a haystack' problem. Intuitively, finding the optimal solution in such high-dimensional spaces seems extremely challenging, but this perception does not always hold in practice. Many scenarios reveal beneficial structures within Lipschitz bandits, which is why our research emphasizes not only worst-case regret but also instance-dependent regret. Our proposed algorithm achieves nearly optimal time and space complexity for both worst-case and instance-dependent regrets.

In practical applications, Lipschitz bandit problems are found in areas such as non-parametric estimation, model selection in machine learning tasks, and decision-making processes in robotics and games. Furthermore, research on Lipschitz bandits has inspired algorithmic advancements in other domains, such as decision trees and tree-based methods, where the principles from Lipschitz bandit algorithms guide the splitting and growth of trees. Despite these advancements, certain limitations remain. High-dimensional Lipschitz bandits can still pose significant computational challenges, especially in cases where the underlying structure is less apparent or more complex. Additionally, the requirement for sufficient exploration to accurately estimate the optimal arm can lead to increased computational overhead in large action spaces.

Our algorithm introduces a novel framework that efficiently addresses online decision-making and balances exploration and exploitation in Lipschitz action spaces. This framework leverages beneficial structures in the problem space to enhance performance while maintaining computational efficiency. We hope our approach could make a substantial contribution to the community, especially in areas that require efficient and effective decision-making under uncertainty.

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

## A Algorithm Flowchart

### A.1 Flowchart for the MBUD Algorithm

The flowchart illustrates the process of the Memory Bounded Uniform Discretization (MBUD) algorithm, showcasing its core steps and transitions. The algorithm begins by dividing the exploration phase into $\lceil \log \log T \rceil$ phases, further segmented into cross exploration phases and the summarize phase.

At the start of the algorithm, the arm space $\mathcal{X} = [0,1]^d$, time horizon $T$, and parameter $c$ are initialized. The initial values for variables such as the best-estimated arm $\boldsymbol{y}$, its average reward $\bar{r}_y$, and the number of pulls $n_y$ are set to zero. The budget parameter $B_{-1}$ is initialized to 1, and the discretization parameter $\epsilon$ is calculated. Rather than covering the entire space continuously, the MBUD algorithm treats subcubes as a stream, selecting cubes for pairwise comparisons and gradually converging to the optimal region.

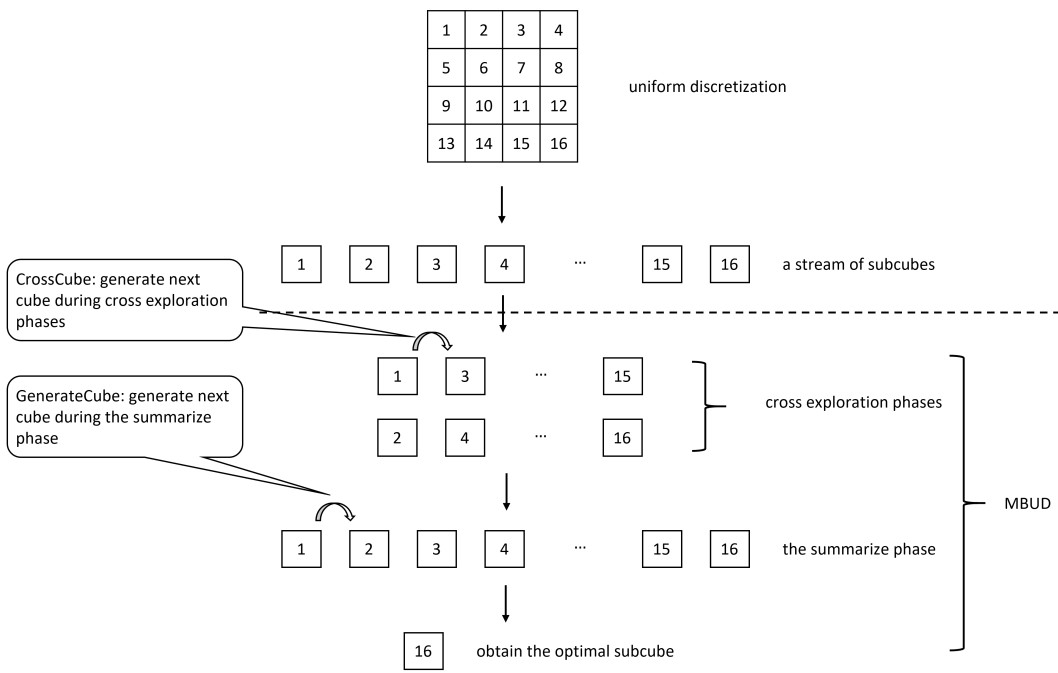

Figure 2: Flowchart for the MBUD algorithm

During the cross exploration phases, which encompass the first $\lceil \log \log T \rceil - 1$ phases, the algorithm iterates over arms within the discretized action space. In each phase $p$, the budget parameter $B_p$ is updated to $\sqrt{TB_{p-1}}$. For each $q$ from 1 to $\lfloor \phi \epsilon^{-d} \rfloor$, the CROSSCUBE function generates a new cube $C$ by calculating parameters $\kappa_1$ and $\kappa_2$ and determining the cube's position using the edge-length $\epsilon$. An arm $\boldsymbol{x}$ is then selected from the cube $C$. The COMPARE function evaluates the selected arm against the current best-estimated arm $\boldsymbol{y}$, updating $\boldsymbol{y}$ if necessary based on the comparison of their upper and lower confidence bounds. In the final phase, known as the summarize phase, the algorithm revisits all arms within the uniform discretization space. For each $q$ from 1 to $\lfloor \epsilon^{-d} \rfloor$, the GENERATECUBE function generates a new cube $C$ without considering the phases, using a parameter $\kappa$ to determine the cube's position. An arm $\boldsymbol{x}$ is selected from this cube and compared against the current best-estimated arm $\boldsymbol{y}$ using the COMPARE function, ensuring thorough evaluation. The algorithm culminates by selecting the best-estimated arm $\boldsymbol{y}$ and playing it for the remaining rounds until the end of the time horizon.

## A.2 Flowchart for the MBAD Algorithm

The MBAD algorithm dynamically adapts its discretization of the action space, focusing more on promising regions to identify the optimal arm with high probability. The flowchart effectively demonstrates how the algorithm narrows down the search space through adaptive discretization. Initially, the algorithm sets up the necessary parameters and variables. During the cross-exploration phases, the ADAPTIVECUBE function generates and selects arms from cubes. Arrows indicate the process of moving to the next cube if $\boldsymbol{y}$ remains the best arm, and the selection and evaluation of subcubes when the comparison budget condition is met.

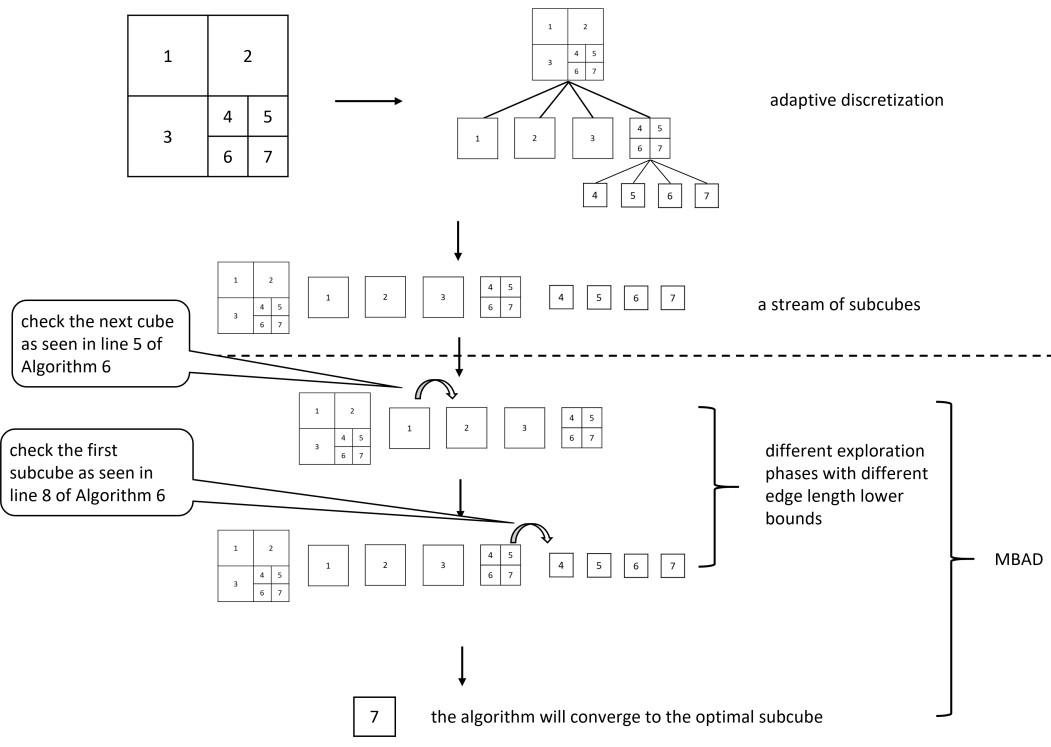

Figure 3: Flowchart for the MBAD algorithm

We present the pseudocode. The algorithm begins by initializing key parameters: the time horizon $T$ and a constant $c$. Initial values for essential variables include the best-estimated arm $\boldsymbol{y}$, its average reward $\bar{r}_y$, and the number of pulls $n_y$, all set to zero. The initial budget $B_1$ is set to $\sqrt{T}$. In each phase $p$, the algorithm initializes a new arm $\boldsymbol{x}$ with zero values for its index, average reward $\bar{r}_x$, and the number of pulls $n_x$. The budget for the current phase $b_p$ is calculated as $B_p \cdot \left(\frac{\log T}{T}\right)^{1/(d+2)}$. The ADAPTIVECUBE function is then called with parameters $m = 4$ and $q = 1$, and the budget for the next phase $B_{p+1}$ is updated to $B_p \log T$.

The ADAPTIVECUBE function is crucial for refining the discretization of the action space and selecting promising arms. It begins by setting the edge-length $\epsilon = 2^{-m}$. The function calculates the parameter $\kappa$ as the largest integer such that $k^d \leq \lfloor \epsilon^{-d} \rfloor$. A node is generated using the geometric sequence $\mathcal{G}_d(q, \kappa)$, and the cube $C$ is defined by this node and $\epsilon$. An arm $\boldsymbol{x}$ is selected from the cube $C$. If $q + 1 \leq 2^m$ and the COMPARE function indicates that $\boldsymbol{y}$ remains the best arm after comparison, the algorithm proceeds to the next cube with parameters $m$ and $q + 1$. If the comparison budget $20\epsilon^{-2}$ does not exceed $b_p$, the algorithm updates $\boldsymbol{y}$ using the COMPARE function, partitions the cube $C$ into $2^d$ subcubes, and checks the first subcube.

## B  Proof of Theorem 1

Let $\mathcal{S}$ denote the $\epsilon$-mesh of the action space where $\epsilon = \left(\frac{\log T}{T}\right)^{1/(d+2)}$. Similarly, the discretization error is the gap of the best fixed arm benchmarks between two spaces:

$$\mathbb{D}_{\mathcal{X}}(\mathcal{S}) = T \cdot \sup_{x \in \mathcal{X}} \mu(x) - T \cdot \sup_{x \in \mathcal{S}} \mu(x).$$

Then the regret could be rewrote as

$$\mathbb{R}_{\mathcal{X}}(T) = \mathbb{R}_{\mathcal{X}}(T) + \mathbb{D}_{\mathcal{X}}(\mathcal{S}).$$

We call $\mathbb{D}_{\mathcal{X}}(\mathcal{S})$ the discretization error and it could be bounded by

$$\mathbb{D}_{\mathcal{X}}(\mathcal{S}) \leq T\epsilon \leq T^{\frac{d+1}{d+2}}(\log T)^{\frac{1}{d+2}}.$$

In the rest of this subsection, we shall prove

$$\mathbb{R}_{\mathcal{S}}(T) \leq \tilde{O}\left(T^{\frac{d+1}{d+2}}\right).$$

For any fixed arm $x \in \mathcal{S}$, with probability $1 - T^{-c}$:

$$|\mu(x) - \bar{r}_x| \leq \sqrt{\frac{c \log T}{n_x}}. \tag{1}$$

By a union bound for all arms and all rounds, (1) holds for all arm $x_t \in \mathcal{S}, t \in [T]$ with probability at least $1 - T^{4-c}$. To ease the reading, we assume $c = 5$. We call this 'clean event' and let $\mathcal{E}$ denote it. Then we analyze the regret based on the clean event. Let $\mathbb{R}_{\mathcal{S}}^p$ denote the regret for the $p$-th phase. Consider $\mathbb{R}_{\mathcal{S}}^0$, because the number of pulls of all arms in phase 0 is bounded by $\epsilon B_0$, we have

$$\mathbb{R}_{\mathcal{S}}^0 \leq 2\sqrt{T}/\log\log T.$$

Then we consider $\mathbb{R}_{\mathcal{S}}^p, p \in [\phi - 1]$. Let $\mu_{\mathcal{S}}^* := \sup_{x \in \mathcal{S}} \mu(x)$ denote the expected per-round reward of the optimal arm in space $\mathcal{S}$. Let $x_p^*$ and $\mu_p^*$ denote the optimal selected arm during phase $p$ and its expected per-round reward, respectively. From the definition of uniform discretization and Lipschitz condition, we have

$$\mu_{\mathcal{S}}^* - \mu_p^* \leq \left(\frac{\log T}{T}\right)^{1/(d+2)} \log\log T. \tag{2}$$

for all phase $p \in [\phi - 1]$. To ease the reading, define

$$\Phi := \left(\frac{\log T}{T}\right)^{1/(d+2)} \log\log T.$$

For phase $p$, we consider the best estimate arm $y$ at the start of the $p$-th phase. If $x_{p-1}^*$ is discarded in phase $p - 1$, according to the stop condition of compare strategy, we have

$$\bar{r}_y \geq \mu_{p-1}^* - \sqrt{\frac{5 \log T}{n_y}} - \sqrt{\frac{5 \log T}{n_{x_{p-1}^*}}} \geq \mu_{p-1}^* - 2\sqrt{\frac{5 \log T}{\epsilon B_{p-1}}}. \tag{3}$$

For arbitrary discarded arm $x$, let $R_x^p$ and $N_x^p$ denote the accumulated reward and total number of pulls during phase $p$, respectively. Notice that the value of $\bar{r}_y - \sqrt{(5 \log T)/n_y}$ is non-decreasing, so we have

$$\frac{R_x^p}{N_x^p - 1} + \sqrt{\frac{5 \log T}{N_x^p - 1}} \geq \mu_{p-1}^* - 2\sqrt{\frac{5 \log T}{\epsilon B_{p-1}}}.$$

Combine (2) and (3) together, we get

$$R_x^p \geq 2N_x^p\left(\mu_{p-1}^* - \sqrt{\frac{5 \log T}{N_x^p - 1}} - \sqrt{\frac{5 \log T}{\epsilon B_{p-1}}}\right).$$

Let $\mathbb{R}^p_x$ denote the cumulative regret of playing arm $x$ during phase $p$, we have

$$\mathbb{R}^p_x \leq 2N^p_x \left( \sqrt{\frac{5\log T}{N^p_x - 1}} + \sqrt{\frac{5\log T}{\epsilon B_{p-1}}} + \Phi \right)$$

$$\leq 2 \left( \sqrt{6N^p_x \log T} + N^p_x \sqrt{\frac{5\log T}{\epsilon B_{p-1}}} + N^p_x \Phi \right).$$

The first term from the gap between the expected reward of best estimated arm and the selected sub-optimal arm. The second term from the deviation between the best estimated arm and optimal expected per-round reward of the $(p-1)$-th phase. And the last is the discretization error during phase $p-1$. Let $\mathcal{S}_p$ denote the set of arms in phase $p$. According to Jensen's inequality

$$\frac{1}{|\mathcal{S}_p|} \sum_{x \in \mathcal{S}_p} \sqrt{N^p_x} \leq \sqrt{\frac{1}{|\mathcal{S}_p|} \sum_{x \in \mathcal{S}_p} N^p_x} \leq \sqrt{\epsilon B_p}.$$

Then we obtain

$$\sum_{x \in \mathcal{S}_p} \sqrt{N^p_x} \leq |\mathcal{S}_p|\sqrt{\epsilon B_p} \leq \sqrt{\frac{B_p}{\epsilon \log\log T}}.$$

Consider all selected arms during phase $p$ and the stop condition of the compare strategy, we have

$$\mathbb{R}^p_{\mathcal{S}} \leq 2 \sum_{x \in \mathcal{S}_p} \left( \sqrt{6N^p_x \log T} + N^p_x \sqrt{\frac{5\log T}{\epsilon B_{p-1}}} + N^p_x \Phi \right)$$

$$\leq \frac{3B_p}{\log\log T} \left( \sqrt{\frac{5\log T}{\epsilon B_{p-1}}} + \Phi \right) + 2\sqrt{6\log T} \sum_{x \in \mathcal{S}_p} \sqrt{N^p_x}$$

$$\leq \frac{3B_p}{\log\log T} \left( \sqrt{\frac{5\log T}{\epsilon B_{p-1}}} + \Phi \right) + 3\sqrt{\frac{6B_p \log T}{\epsilon \log\log T}}.$$

For the incurred regret by the deviation between the expected reward of best estimated arm and the selected sub-optimal arm, we have

$$\sum_{p=1}^{\phi-1} 3\sqrt{\frac{6B_p \log T}{\epsilon \log\log T}} \leq 6\sqrt{\frac{6B_{\phi-1} \log T}{\epsilon \log\log T}} \leq 6\sqrt{\frac{6T \log T}{\epsilon \log\log T}} \leq \tilde{O}\left( T^{\frac{d+1}{d+2}} \right).$$

For the incurred regret the deviation between the best estimated arm and optimal expected per-round reward, we have

$$\sum_{p=1}^{\phi-1} \frac{3B_p}{\log\log T} \sqrt{\frac{5\log T}{\epsilon B_{p-1}}} \leq 6B_{\phi-1}\sqrt{\frac{5\log T}{\epsilon B_{\phi-2}}} \leq 6\sqrt{T}\sqrt{\frac{5\log T}{\epsilon}}.$$

For the incurred regret of the discretization error during one phase, we obtain

$$\sum_{p=1}^{\phi-1} \frac{3B_p \Phi}{\log\log T} \leq \frac{6B_{\phi-1}\Phi}{\log\log T} \leq \tilde{O}\left( T^{\frac{d+1}{d+2}} \right).$$

Combine them together, we get

$$\sum_{p=1}^{\phi-1} \mathbb{R}^p_{\mathcal{S}} \leq \tilde{O}\left( T^{\frac{d+1}{d+2}} \right).$$

Then we consider the total cumulative regret during the exploration part. Let $\mathbb{R}^{\phi}_{\mathcal{S}}$ denote the regret incurred in the last part. According to that the value of $\bar{r}_y - \sqrt{(5\log T)/n_y}$ is non-decreasing and the relationship between $B_\phi$ and $B_{\phi-1}$, we have

$$\mathbb{R}^{\phi}_{\mathcal{S}} \leq \mathbb{R}^{\phi-1}_{\mathcal{S}}.$$

573 Then the regret incurred by the exploration is

$$\sum_{p=0}^{\phi} \mathbb{R}_{\mathcal{S}}^p \leq 2 \sum_{p=0}^{\phi-1} \mathbb{R}_{\mathcal{S}}^p \leq \tilde{O}\left(T^{\frac{d+1}{d+2}}\right).$$

574 Consider the selected arm $y$ after the exploration and let $\mathbb{R}_{\mathcal{S}}^y$ denote the regret due to selecting it.
575 According to the stop condition of compare strategy, we have

$$\bar{r}_y \geq \mu_{\mathcal{S}}^* - 2\sqrt{\frac{5 \log T}{\epsilon B_{\phi-1}}}.$$

576 Then for the regret

$$\mathbb{R}_{\mathcal{S}}^y \leq 2T \sqrt{\frac{5 \log T}{\epsilon B_{\phi-1}}} \leq \tilde{O}\left(T^{\frac{d+1}{d+2}}\right).$$

577 Based on the clean event, we have

$$\mathbb{E}[\mathbb{R}_{\mathcal{S}}(T)|\mathcal{E}] = \mathbb{R}_{\mathcal{S}}^y + \sum_{p=0}^{\phi} \mathbb{R}_{\mathcal{S}}^p \leq \tilde{O}\left(T^{\frac{d+1}{d+2}}\right).$$

578 Then the regret is

$$\begin{aligned}
\mathbb{R}_{\mathcal{S}}(T) &= \mathbb{E}[\mathbb{R}_{\mathcal{S}}(T)|\mathcal{E}] \cdot \mathbb{P}(\mathcal{E}) + \mathbb{E}[\mathbb{R}_{\mathcal{S}}(T)|\neg\mathcal{E}] \cdot \mathbb{P}(\neg\mathcal{E}) \\
&\leq [\tilde{O}\left(T^{\frac{d+1}{d+2}}\right)](1 - 1/T) + 1 \\
&\leq \tilde{O}\left(T^{\frac{d+1}{d+2}}\right).
\end{aligned}$$

579 Combine it with the discretization error, then we complete the proof.

## C Proof of Theorem 2

581 To ease the reading, let $c = 5$. For all arms $x_t \in \mathcal{X}$ and all rounds $t \in [T]$, the gap between the mean
582 reward and the expectation could be bounded with probability $1 - T^{-1}$:

$$|\mu(x_t) - \bar{r}_{x_t}| \leq \sqrt{\frac{5 \log T}{n_{x_t}}}, \forall t \in [T].$$

583 We call this 'clean event' $\mathcal{E}$ and mainly analyze the regret based on $\mathcal{E}$. Assume the MBAD algorithm
584 consume all time horizons during the $\phi$-th phase. For the stochastic Lipschitz instance, we always
585 have $\phi \leq O\left(\frac{\log T}{\log \log T}\right)$. Let $\mathbb{R}_{\mathcal{S}}^p$ denote the regret for the $p$-th phase. For the first phase, we have
586 $\mathbb{R}_{\mathcal{X}}^1 \leq N^1 \leq B_1 \leq \sqrt{T}$. Then we consider $\mathbb{R}_{\mathcal{S}}^p, 1 < p \leq \phi$. For phase $p$, we consider the best
587 estimate arm $y$ at the start of the $p$-th phase. If $x_{p-1}^*$ is discarded in phase $p - 1$, according to the
588 stop condition of compare strategy, we have

$$\bar{r}_y \geq \mu_{p-1}^* - \sqrt{\frac{5 \log T}{n_y}} - \sqrt{\frac{5 \log T}{n_{x_{p-1}^*}}} \geq \mu_{p-1}^* - 2\sqrt{\frac{5 \log T}{b_{p-1}}}.$$

589 For arbitrary discarded arm $x$, let $R_x^p$ and $N_x^p$ denote the accumulated reward and total number of
590 pulls during phase $p$, respectively. Notice that the value of $\bar{r}_y - \sqrt{(5 \log T)/n_y}$ is non-decreasing, so
591 we have

$$\frac{R_x^p}{N_x^p - 1} + \sqrt{\frac{5 \log T}{N_x^p - 1}} \geq \mu_{p-1}^* - 2\sqrt{\frac{5 \log T}{b_{p-1}}}.$$

592 Then we get

$$R_x^p \geq 2N_x^p \left(\mu_{p-1}^* - \sqrt{\frac{5 \log T}{N_x^p - 1}} - \sqrt{\frac{5 \log T}{b_{p-1}}}\right).$$

593 Let $\mathbb{R}_x^p$ denote the cumulative regret of playing arm $x$ during phase $p$, we have

$$\mathbb{R}_x^p \leq 2N_x^p \left( \sqrt{\frac{5\log T}{N_x^p - 1}} + \sqrt{\frac{5\log T}{b_{p-1}}} \right).$$

594 Similarly, the first term from the gap between the expected reward of best estimated arm and the
595 selected sub-optimal arm. The second term from the deviation between the best estimated arm and
596 optimal expected per-round reward of the $(p-1)$-th phase. Recall the set

$$\mathcal{Y}_i = \{x \in X : 2^{-i} \leq \Delta(x) < 2^{1-i}, i \in \mathbb{N}\},$$

597 and the definition of zooming dimension

$$d_z = \inf_{\beta \geq 0} \left\{ |\mathcal{S}_j| \leq O(\epsilon^\beta), \epsilon = O(2^{-j}), \forall j \in \mathbb{N} \right\}.$$

598 Pick $\delta = \left( \frac{\log^2 T}{T} \right)^{\frac{1}{d_z+2}}$, if $\sqrt{\frac{5\log T}{N_x^p-1}} + \sqrt{\frac{5\log T}{b_{p-1}}} \leq O(\delta)$, then $\mathbb{R}_x^p \leq O(\delta N_x^p)$. If $\sqrt{\frac{5\log T}{b_{p-1}}} > \Omega(\delta)$,
599 then $b_{p-1} = O(\log T)\Delta^{-2}(x)$. If $\sqrt{\frac{5\log T}{N_x^p-1}} > \Omega(\delta)$, then $N_x^p = O(\log T)\Delta^{-2}(x)$. According the
600 stop condition of the compare strategy and the definition of zooming dimension, we have

$$\begin{aligned}
\mathbb{R}_\mathcal{X}^p &\leq \delta N^p + \sum_{i:2^{-i} > \delta} \sum_{x \in Y_i} \mathbb{R}_x^p \\
&\leq \delta N^p + O((\log T)^2)\delta^{d_z+1} \leq \delta T + O((\log T)^2)\delta^{d_z+1} \\
&\leq O(T^{\frac{d_z+1}{d_z+2}}(\log T)^{\frac{2}{d_z+2}}).
\end{aligned}$$

601 Then we have

$$\sum_{p=1}^{\phi} \mathbb{R}_\mathcal{X}^p \leq \sum_{p=1}^{\phi} O(T^{\frac{d_z+1}{d_z+2}}(\log T)^{\frac{2}{d_z+2}}) \leq \tilde{O}(T^{\frac{d_z+1}{d_z+2}}).$$

602 Based on the clean event, we have

$$\mathbb{E}[\mathbb{R}_\mathcal{X}(T)|\mathcal{E}] \leq \sum_{p=1}^{\phi} \mathbb{R}_\mathcal{X}^p \leq \tilde{O}(T^{\frac{d_z+1}{d_z+2}}).$$

603 The regret is

$$\mathbb{R}_\mathcal{X}(T) \leq \tilde{O}(T^{\frac{d_z+1}{d_z+2}})(1 - 1/T) + 1 \leq \tilde{O}(T^{\frac{d_z+1}{d_z+2}}).$$

604 Then we complete the proof.

## D A Simple Algorithm

606 We present the pseudocode. The algorithm also maintains the index $x$, the mean reward estimator
607 $\bar{r}_x$, and the number of pulls $n_x$ for one arm in memory. The constants $c$ and $\eta$ are two parameters
608 to balance the exploration and exploitation. For each phase $p$, the algorithm determines the budget
609 $b_p$ of samples for each probe, the number of total pulls $N^p$ during the phase, and the usage factor
610 $\lambda_p$ obtained after the phase. Once the arm $x$ is discarded, the algorithm chooses the next probe by
611 adding $\eta\sqrt{(c\log T)/(n_x L^2)}$, which is the step size and has the same order as the confidence radius
612 of arm $x$.

613 We have the following theoretical result.

614 **Theorem 3.** *For Lipschitz bandits with time horizon $T$ and Lipschitz constant $L$, Algorithm 7 with*
615 *$c \geq 5$ and $\eta = 1/3$ achieves regret*

$$\mathbb{R}_\mathcal{X}(T) \leq \tilde{O}(T^{\frac{d_z+1}{d_z+2}}),$$

616 *using $O(1)$ stored arms, where $d_z \leq 1$ is the zooming dimension of space $\mathcal{X}$.*

**Algorithm 7** A Simple Algorithm

**Input:** rounds $T$, Lipschitz constant $L$, constant $c$ and $\eta$

1: $y \leftarrow 0, \bar{r}_y \leftarrow 0, n_y \leftarrow 0, B_1 \leftarrow \sqrt{T}$.
2: **for** $p = 1, 2, \cdots$ **do**
3: $\quad$ $x \leftarrow 0, N^p \leftarrow 0$.
4: $\quad$ **while** $x \leq 1$ **do**
5: $\quad\quad$ $\bar{r}_x \leftarrow 0, n_x \leftarrow 0, b_p \leftarrow (B_p/3)^{2/3}(c \log T)^{1/3} L^{-2/3}$.
6: $\quad\quad$ **while** $n_x \leq b_p$ or $n_y \leq b_p$ **do**
7: $\quad\quad\quad$ $N^p \leftarrow N^p + 1$.
8: $\quad\quad\quad$ Pull the least played arm between $x$ and $y$, and select a random arm if there not exists a least played arm.
9: $\quad\quad\quad$ Update $\bar{r}_x, n_x, \bar{r}_y, n_y$.
10: $\quad\quad\quad$ **if** $\min\{\bar{r}_x + \sqrt{(c \log T)/n_x}, 1\} < \max\{\bar{r}_y - \sqrt{(c \log T)/n_y}, 0\}$ **then**
11: $\quad\quad\quad\quad$ Break.
12: $\quad\quad\quad$ **else if** $\max\{\bar{r}_x - \sqrt{(c \log T)/n_x}, 0\} > \max\{\bar{r}_y - \sqrt{(c \log T)/n_y}, 0\}$ **then**
13: $\quad\quad\quad\quad$ $y \leftarrow x, \bar{r}_y \leftarrow \bar{r}_x, n_y \leftarrow n_x$.
14: $\quad\quad\quad\quad$ Break.
15: $\quad\quad\quad$ **end if**
16: $\quad\quad$ **end while**
17: $\quad\quad$ $x \leftarrow x + \eta\sqrt{(c \log T)/(n_x L^2)}$.
18: $\quad$ **end while**
19: $\quad$ $B_{p+1} \leftarrow B_p \log T$.
20: **end for**

### D.1 Proof of Theorem 3

The proof closely mirrors that of Theorem 2. To ease the reading, let $c = 5$ and $\eta = 1/3$. For all arms $x_t \in \mathcal{X}$ and all rounds $t \in [T]$, the gap between the mean reward and the expectation could be bounded with probability $1 - T^{-1}$:

$$|\mu(x_t) - \bar{r}_{x_t}| \leq \sqrt{\frac{5 \log T}{n_{x_t}}}, \forall t \in [T].$$

We call this 'clean event' $\mathcal{E}$ and mainly analyze the regret based on $\mathcal{E}$. For the number of phases $\phi$, we always have $\phi \leq O\left(\frac{\log T}{\log \log T}\right)$. Notice that the number of total pulls during the phase $p$,

$$N^p \leq 3b_p L \sqrt{\frac{b_p}{5 \log T}} \leq 3(B_p/3)^{2/3}(5 \log T)^{1/3}\sqrt{\frac{(B_p/3)^{2/3}(5 \log T)^{1/3}}{5 \log T}} = B_p.$$

Let $\mathbb{R}_S^p$ denote the regret for the $p$-th phase. For the first phase, we have $\mathbb{R}_\mathcal{X}^1 \leq N^1 \leq B_1 \leq \sqrt{T}$. Then we consider $\mathbb{R}_S^p, 1 < p \leq \phi$. For phase $p$, we consider the best estimate arm $y$ at the start of the $p$-th phase. If $x_{p-1}^*$ is discarded in phase $p-1$, according to the stop condition of compare strategy, we have

$$\bar{r}_y \geq \mu_{p-1}^* - \sqrt{\frac{5 \log T}{n_y}} - \sqrt{\frac{5 \log T}{n_{x_{p-1}^*}}} \geq \mu_{p-1}^* - 2\sqrt{\frac{5 \log T}{b_{p-1}}}.$$

For arbitrary discarded arm $x$, let $R_x^p$ and $N_x^p$ denote the accumulated reward and total number of pulls during phase $p$, respectively. Notice that the value of $\bar{r}_y - \sqrt{(5 \log T)/n_y}$ is non-decreasing, so we have

$$\frac{R_x^p}{N_x^p - 1} + \sqrt{\frac{5 \log T}{N_x^p - 1}} \geq \mu_{p-1}^* - 2\sqrt{\frac{5 \log T}{b_{p-1}}}.$$

Then we get

$$R_x^p \geq 2N_x^p \left(\mu_{p-1}^* - \sqrt{\frac{5 \log T}{N_x^p - 1}} - \sqrt{\frac{5 \log T}{b_{p-1}}}\right).$$

Let $\mathbb{R}_x^p$ denote the cumulative regret of playing arm $x$ during phase $p$, we have

$$\mathbb{R}_x^p \leq 2N_x^p \left( \sqrt{\frac{5\log T}{N_x^p - 1}} + \sqrt{\frac{5\log T}{b_{p-1}}} \right).$$

Similarly, the first term from the gap between the expected reward of best estimated arm and the selected sub-optimal arm. The second term from the deviation between the best estimated arm and optimal expected per-round reward of the $(p-1)$-th phase. Recall the set

$$\mathcal{Y}_i = \{x \in X : 2^{-i} \leq \Delta(x) < 2^{1-i}, i \in \mathbb{N}\},$$

and the definition of zooming dimension

$$d_z = \inf_{\beta \geq 0} \left\{ |\mathcal{S}_j| \leq O(\epsilon^\beta), \epsilon = O(2^{-j}), \forall j \in \mathbb{N} \right\}.$$

Pick $\delta = \left( \frac{\log^2 T}{T} \right)^{\frac{1}{d_z+2}}$, if $\sqrt{\frac{5\log T}{N_x^p-1}} + \sqrt{\frac{5\log T}{b_{p-1}}} \leq O(\delta)$, then $\mathbb{R}_x^p \leq O(\delta N_x^p)$. If $\sqrt{\frac{5\log T}{b_{p-1}}} > \Omega(\delta)$, then $b_{p-1} = O(\log T)\Delta^{-2}(x)$. If $\sqrt{\frac{5\log T}{N_x^p-1}} > \Omega(\delta)$, then $N_x^p = O(\log T)\Delta^{-2}(x)$. According the stop condition of the compare strategy and the definition of zooming dimension, we have

$$\begin{aligned}
\mathbb{R}_\mathcal{X}^p &\leq \delta N^p + \sum_{i:2^{-i}>\delta} \sum_{x \in Y_i} \mathbb{R}_x^p \\
&\leq \delta N^p + O((\log T)^2)\delta^{d_z+1} \leq \delta T + O((\log T)^2)\delta^{d_z+1} \\
&\leq O(T^{\frac{d_z+1}{d_z+2}}(\log T)^{\frac{2}{d_z+2}}).
\end{aligned}$$

Then we have

$$\sum_{p=1}^{\phi} \mathbb{R}_\mathcal{X}^p \leq \sum_{p=1}^{\phi} O(T^{\frac{d_z+1}{d_z+2}}(\log T)^{\frac{2}{d_z+2}}) \leq \tilde{O}(T^{\frac{d_z+1}{d_z+2}}).$$

Based on the clean event, we have

$$\mathbb{E}[\mathbb{R}_\mathcal{X}(T)|\mathcal{E}] \leq \sum_{p=1}^{\phi} \mathbb{R}_\mathcal{X}^p \leq \tilde{O}(T^{\frac{d_z+1}{d_z+2}}).$$

The regret is

$$\mathbb{R}_\mathcal{X}(T) \leq \tilde{O}(T^{\frac{d_z+1}{d_z+2}})(1 - 1/T) + 1 \leq \tilde{O}(T^{\frac{d_z+1}{d_z+2}}).$$

Then we complete the proof.

# E    Numerical Results

**Different variances.**    Keeping other setting of Figure 1(b) unchanged, Figure 4(a-b) present the results with different variances. For $\xi \sim \mathcal{N}(0, 0.05^2)$ (Figure 4(a)), the MBAD algorithm achieves $140.2\%$ regret of the zooming algorithm. For $\xi \sim \mathcal{N}(0, 0.2^2)$ (Figure 4(b)), the MBAD algorithm achieves $149.5\%$ regret of the zooming algorithm. Overall, our algorithm performs better when the variance is small. Note that the algorithm is based on the 'successive elimination-style' strategy and smaller variances make the algorithm select better arms during comparisons with higher probability.

**Uniform noise distribution.**    Keeping other setting of Figure 1(b) unchanged, Figure 5(a) presents the results with uniform noise distribution. For $\xi \sim \mathcal{U}(-0.2, 0.2)$ (Figure 5(a)), the MBAD algorithm achieves $118.1\%$ regret of the zooming algorithm. The results show that our algorithms work robustly for different noise distributions.

**Quadratic reward function.**    We also provide the numerical results for different reward functions. Keeping other setting of Figure 1(b) unchanged, Figure 5(b) presents the results with uniform noise distribution. For $f(x) = 1 - 4 \times (0.5 - x)^2$ (Figure 5(b)), the MBAD algorithm achieves $132.3\%$ regret of the zooming algorithm. The results show that our algorithms work robustly for different reward functions.

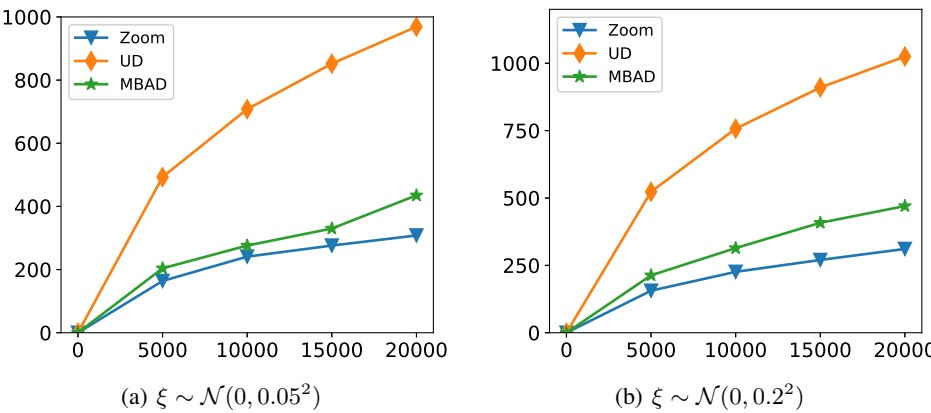

Figure 4: Performance comparisons for Gaussian distribution with different variances.

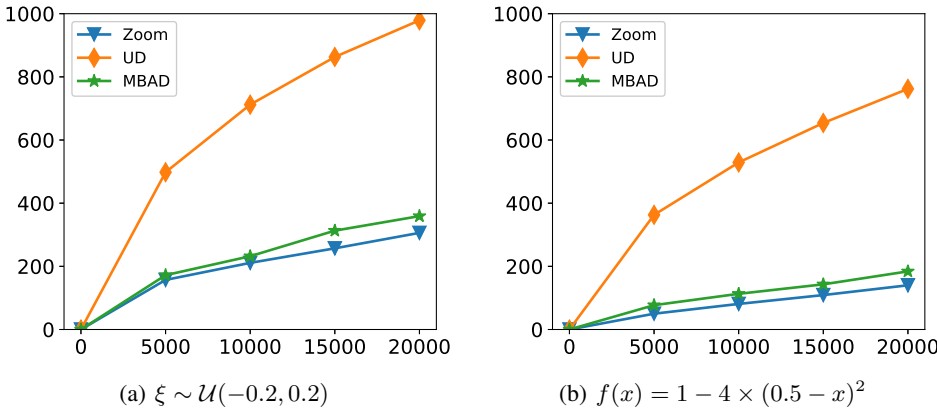

Figure 5: Performance comparisons for (a) Uniform distribution; (b) Quadratic reward function.

