# OpenReview forum: "Efficient Algorithms for Lipschitz Bandits"
_NeurIPS.cc/2024/Conference — Submitted to NeurIPS 2024_

### Official Review · Reviewer_Bcg5 · 2024-06-13

**Soundness:** 1
**Presentation:** 1
**Contribution:** 4
**Rating:** 2
**Confidence:** 5

**Summary:**

The paper proposes two algorithms for Lipschitz bandit problems, with improved time complexity and memory requirements.

**Strengths:**

If the algorithms and proofs are sound, then this is an excellent contribution. Developing these sort of streaming/sketching methods for key bandit problems (such as Lipschitz bandits) is an important area of research, and many people are likely to care about the results of this paper.

**Weaknesses:**

The paper is sloppy to the extent that it is difficult to understand the authors' algorithm or verify their claims. To be specific, consider the following sentences, all taken from a single two-paragraph subsection (section 2.2):
1. "Let $\\{\mathcal{X}_1, \dotsc, \mathcal{X}_N\\}[\mathcal{X}_i \subset \mathcal{X}]$ be an cover of the action space $\mathcal{X}$" --- okay, what is the $\\{\dotsc\\}[\dotsc]$ notation?
2. "Let $\epsilon$ denote the maximum diameter of $\mathcal{X}_i$ for all $i \in [N]$." --- okay, but what is a diameter? Are we in a metric space? This hasn't been specified.
3. "Then the arm set $S = \\{ x_i \mid x_i \in \mathcal{X}_i, i \in [N]\\}$ is an $\epsilon$-mesh."  --- what set is this? Now, I presume that the authors mean to say that they want $S$ to be any set that contains a single element chosen arbitrarily from each of the $\mathcal{X}_i$, but that's not written, instead the authors said its __the__ set, but the right hand side does not specify any unique set. Also, the concept of an $\epsilon$-mesh has not been defined, and when its defined, it needs to be with respect to some metric. And if this description was meant to be the definition of an $\epsilon$-mesh... then that's not clear either (and the definition given by the work the authors state these definitions are from, i.e. Slivkins 2019, is _very_ clear---all the authors needed to do was copy it).
4. "The covering dimension $d$ of the action space $\mathcal{X}$ is defined as $d=\inf_{\alpha \geq 0}\\{|\mathcal{S}| \leq \epsilon^{-\alpha}, \forall \epsilon > 0\\}$." But the set $\mathcal{S}$ does not depend on $\epsilon$ (not even implicitly)... (the correct definition, I presume, would be to ask that $\mathcal{S}\_{\epsilon}$ is a minimal $\epsilon$-cover of $\mathcal{X}$ in some metric $D$, and then have that infimum include $\mathcal{S}_\epsilon$ and not $\mathcal{S}$.)
5. "Define $\mathcal{Y}\_j = \\{x \in \mathcal{X} \colon 2^{-j} \leq \Delta(x) \leq 2^{1-j}, j \in \mathbb{N}\\}$, then the set $\mathcal{Y}_j$ contains all arms whose gap is between $2^{-j}$ and $2^{1-j}$." --- but $j \in \mathbb{N}$ is within the constructions of the set on the RHS, which could be read as asking that the condition holds for all such $j$, or for some $j$, but it breaks the dependence of the right hand side on the subscript $j$ of $\mathcal{Y}_j$. Of course, the definition shouldn't have the $j \in \mathbb{N}$ inside the $\\{ \dotsc \\}$ on the right hand side of the definition.
6. "Consider the $\epsilon$-mesh $\mathcal{S}_j$ for space $\mathcal{Y}_j$." --- __the__ $\epsilon$-mesh? Also, $\mathcal{S}_j$ hasn't been defined. This should say instead 'fix some $\epsilon > 0$ and let $\mathcal{S}_j$ be an $\epsilon$-mesh of $\mathcal{Y}_j$', or something like that.
7. "[...] the zooming dimension focuses only on the set $\mathcal{Y}_j$" --- no, the zooming dimension depends on all the sets $\mathcal{Y}_1, \mathcal{Y}_2, \dotsc$, not only a single one of those sets.

While each individual mistake or ambiguity can be resolved easily enough, verifying the authors claims would require me to rewrite everything myself, and this goes beyond what I'm willing to do (and should do...).  The whole paper is like this, and it's just not acceptable.

I would urge the authors to, in the future, have someone _not intimately familiar with the work_ proof-read the work.

Note, I put down confidence as "5: You are absolutely certain about your assessment. You are very familiar with the related work and checked the math/other details carefully." --- I am indeed very familiar with the related work, but I have not checked the math/other details carefully. It's too much work to read it. I am absolutely certain, however, that this level falls short of any level of clarity that might be expected in published work.

**Questions:**

.

**Limitations:**

.

---

> ### Author Rebuttal · Authors · 2024-08-07
>
> We sincerely appreciate your valuable time and effort in reviewing this paper. We thank you for the thoughtful feedback you provided, which has significantly improved the quality of this paper. We also appreciate your recognition of the contributions our paper makes.
>
> In Section 2.2, we primarily introduce the concepts of Covering Dimension and Zooming Dimension to provide background for readers who may be unfamiliar with them. We acknowledge that due to page limitations, some concepts may not have been clearly or comprehensively defined. We will revise this section based on your feedback. However, it is worth noting that even if this section were entirely omitted, it would not affect the paper's overall framework and contributions. While this section is not essential to the core of the paper, we appreciate your attention and suggestions regarding it. We hope you will consider the other sections of the paper to comprehensively evaluate the paper's contributions and value.
>
> Please feel free to reach out if there are any other aspects you would like us to address or discuss further. We are more than happy to engage in further dialogue to ensure all concerns are comprehensively addressed.

---

> ### Comment · Reviewer_Bcg5 · 2024-08-08
>
> To clarify, the issue isn't section 2.2. That is just an example. This level of "messiness" is present throughout.
>
> This isn't a matter of the definitions being hard to understand due to a lack of space: clear definitions would not take more space at all.
>
> And, on that point, if, as you claim, you could remove section 2.2 from the paper, and still fully understand the contribution... why is that section in your paper?
>
> __To area chair:__ I fully stand by my recommendation to reject this work. I'd like to point out that the other reviewer's comments were:
> 1. Reviewer q7Fp:  "I did have a little trouble reading the parts about the crosscut and generating cubes but it might just be me not being familiar with prior work in the area."
> 2. Reviewer svqm: "It is not simple to parse algorithms in their current form in a short amount of time. Although you give comprehensive descriptions in text, I believe adding illustrations or additional explanations will significantly improve clarity of your algorithms. "
>
> It's not a matter of Reviewer q7Fp being unfamiliar with prior work---I am familiar with it, and those parts require deciphering, not reading. And its not a matter of Reviewer svqm struggling to parse the algorithms in a short time---the algorithms are not parsable. Additional illustrations are nice, but should not be necessary for a skilled researcher to parse an algorithm. I am certain that if I gave the algorithm blocks in this paper to 5 strong master's students to implement, they would each return with a different algorithm. Reviewer mKhe states that "The paper has good presentation"---I find this wild; I can only presume that the reviewer did not try to understand the details of how the algorithm works, or study the proofs in any depth; then yes, at a high level, the paper might seem reasonable---but this is a theory paper, high level does not suffice.

---

> > ### Author Response · Authors · 2024-08-08
> >
> > Thank you very much for your prompt response and valuable feedback. We appreciate the opportunity to improve our paper through your insights. Below is our reply:
> >
> > 1. **Acknowledgment:** We sincerely appreciate the constructive feedback from each reviewer, which has helped identify aspects of our paper that were unclear. We would like to express our gratitude once again to all the reviewers for their thoughtful comments. In our rebuttal, we have provided further explanations and worked diligently to improve the paper's quality, which we believe is one of the key values and purposes of the rebuttal period.
> >
> > 2. **Presentation Feedback:** While reviewers q7Fp and svqm pointed out specific aspects where the algorithm's presentation could be improved, it is also worth noting that the other three reviewers, including q7Fp and svqm, rated the presentation as "**3 - good**". This suggests that the presentation is generally well-received, though we acknowledge there is always room for refinement to enhance clarity and understanding.
> >
> > 3. **Implementation Variability:** You mentioned that "if I gave the algorithm blocks in this paper to 5 strong master's students to implement, they would each return with a different algorithm". This variability is quite common, as many algorithms can be implemented in multiple ways. For instance, in the Lipschitz bandits area, the well-known Zooming algorithm, when only considering the original paper's description, can be implemented in several different ways.
> >
> > 4. **Section 2.2:** As mentioned in our rebuttal, we introduced the concepts of Covering Dimension and Zooming Dimension primarily to provide background for readers who may be unfamiliar with them. While this section enhances understanding, many papers in the field do not include such introductions. Of course, we will continue to refine this section to make it more accurate and clear, but its presence or absence does not affect the paper's overall framework and contributions.

---

### Official Review · Reviewer_svqm · 2024-07-08

**Soundness:** 3
**Presentation:** 3
**Contribution:** 3
**Rating:** 6
**Confidence:** 3

**Summary:**

The paper considers regret minimization for Lipschitz bandits with time horizont $T$ and proposes an algorithm that provably achieves nearly optimal regret while having strictly smaller (by a factor of $T$) time (of order $O(T)$) and memory complexity (of order $O(1)$). This is achieved by considering a tree-like embedding of the state space and pairwise comparison between elements of the tree. A suboptimal method with uniform discretization called MBUD has dependence on the covering dimension of the state space, while MBAD, a method with adaptive discretization, instead has dependence only on the zooming dimension.

**Strengths:**

1) Achieving nearly optimal regret bounds in minimax (MBUD) and instance-specific setting (MBAD), while reducing time and memory complexity.

2) Both proposed algorithms are non-trivial and seem to be novel and interesting on their own.

**Weaknesses:**

1) It is not simple to parse algorithms in their current form in a short amount of time. Although you give comprehensive descriptions in text, I believe adding illustrations or additional explanations will significantly improve clarity of your algorithms.

2) I would appreciate a more explicit comparison with previous work - what parts of the algorithms were already reported in the literature?

**Questions:**

1) Could you provide intuition behind your node exploration process i.e. line 2 in Algorithm 2?

2) (196) Should each cross exploration phase explore instead $O(\\log \\log T)$ cubes?

3) Could you please refer to previous works that worked on reducing time and memory complexity for bandits, and, more specifically, whether it is a common thing to achieve the same regret bounds for settings with restricted time and memory?

4) Could you please comment what part of the proposed algorithms has already been used before in literature? I am especially interested whether cube generation i.e. the way you do tree search has been used before in this context.

**Limitations:**

The authors have addressed limitations adequately.

---

> ### Author Rebuttal · Authors · 2024-08-07
>
> We sincerely appreciate your valuable time and effort in reviewing this paper. We thank you for the thoughtful feedback you provided, which has significantly improved the quality of this paper. For the potential concerns you bring up, we would like to address them here.
>
> **Q1: It is not simple to parse algorithms in their current form in a short amount of time. Although you give comprehensive descriptions in text, I believe adding illustrations or additional explanations will significantly improve the clarity of your algorithms.**
>
> Thank you for your suggestion. Due to page limitations in the main text, we have included a flowchart in Appendix A of the paper, which visually represents the algorithm's process and main ideas to facilitate reader understanding.
>
> **Q2: I would appreciate a more explicit comparison with previous work - what parts of the algorithms were already reported in the literature?**
>
> The discretization method used in the algorithm is standard and widely used in the field. The novel aspect of our algorithm lies in how we explore these subcubes after discretization. Previous works typically consider all subcubes simultaneously. In contrast, we designed a new exploration process and algorithm that performs pairwise comparisons between these subcubes in each round, thereby reducing both the time and space complexity of the algorithm.
>
> **Q3: Could you provide intuition behind your node exploration process, i.e., line 2 in Algorithm 2?**
>
> We have included a flowchart in Appendix A of the paper to illustrate the intuition behind our approach. Before the final phase, the MBUD algorithm explores all subcubes. Therefore, it is crucial to allocate which subcubes to explore in each phase to maximize exploration efficiency. Our approach involves selecting every few subcubes in each phase, ensuring that most subcubes are explored before the final phase. Line 2 in Algorithm 2 provides the formula for this selection process.
>
> **Q4: (196) Should each cross-exploration phase explore instead $O(\log \log T)$ cubes?**
>
> Thank you for pointing this out. The original sentence, “Each cross-exploration phase will only explore $O\left(\frac{1}{\log \log T}\right)$ of them,” should be revised to “Each cross-exploration phase will only explore approximately $\frac{1}{\log \log T}$ of them.”
>
> **Q5: Could you please refer to previous works that worked on reducing time and memory complexity for bandits, and, more specifically, whether it is a common thing to achieve the same regret bounds for settings with restricted time and memory?**
>
> Typically, algorithms with limited arm storage tend to increase regret, which aligns with our intuition. Therefore, an interesting question is whether the regret gap can be reduced if additional information is gained during the exploration process or if there are connections between arms. For example, in Lipschitz bandits, once the reward range of a subcube is known, we can estimate the reward range of neighboring subcubes, helping us reduce the reward gap between full memory and memory-constrained settings.
>
> **Q6: Could you please comment on what part of the proposed algorithms has already been used before in the literature? I am especially interested in whether cube generation, i.e., the way you do tree search, has been used before in this context.**
>
> Thank you for your question. As we noted in our response to Q2, our exploration method is novel. The approach of cube generation and tree search is new in this context.
>
> **Please kindly let us know if you have any concerns you find not fully addressed. We are more than happy to have a further discussion regarding them.**

---

> ### Comment · Reviewer_svqm · 2024-08-12
>
> Thank you for your response. I was aware of the flowcharts in Appendix A at the time of writing the review, but I did not find them particularly intuitive or explanatory. Considering the concerns raised by Reviewer Bcg5, I agree that the paper could have been written more clearly. I will maintain my score for now and closely follow the discussion surrounding the issues raised by Reviewer Bcg5.

---

### Official Review · Reviewer_mKhe · 2024-07-12

**Soundness:** 4
**Presentation:** 3
**Contribution:** 3
**Rating:** 6
**Confidence:** 3

**Summary:**

The paper investigates a multi-armed bandit problem where the action space is a metric space a stochastic Lipschitz rewards. The authors present algorithms that use a constant amount of memory and achieve a near-optimal regret. This improves on previous results that had heavy memory usage.

**Strengths:**

The paper has good presentation, and the figures in the appendices are helpfulץ The contribution itself is useful in practice.

**Weaknesses:**

While the result is great, the ideas presented in the paper are modifications of existing methods

**Questions:**

* Line 136: Should be norm rather than absolute value, right?
* Line 156: Definition of the mesh seems like it contains ALL elements of a cover set. Should be only one per set?
* Line 196: The use of $O$ notation for the fraction is nonstandard and confusing (as is it looks like $O(1)$)

**Limitations:**

I did not find the limitations presented as sufficient and I would like to see more discussion on the downsides of the presented algorithm and future directions of research.

---

> ### Author Rebuttal · Authors · 2024-08-07
>
> We sincerely appreciate your valuable time and efforts in reviewing this paper. We thank the thoughtful feedback you provided, which significantly improved the quality of this paper. For the potential concerns you bring up, we would like to answer/address them here.
>
> **Q1: "While the result is great, the ideas presented in the paper are modifications of existing methods.**
>
> We thank you for your insightful comment. The novelty of our paper lies primarily in the design of the algorithm and the novel theoretical results we obtained. Regarding the algorithm design, our main contribution can be divided into two key insights.
>
> The first insight involves the use of metric embedding, which maps elements from one metric space to another while preserving distance relationships as much as possible. The critical aspect of metric embedding is finding a mapping that maintains the original space's distance metrics, making data processing, analysis, and understanding more efficient and intuitive. Our algorithm essentially maps the metric space to a tree structure, where each node corresponds to a cube. Traversing the nodes of this tree is equivalent to traversing the entire metric space, allowing for a more structured and efficient exploration.
>
> The second insight leverages pairwise comparisons of arms to reduce memory complexity. Rather than covering the entire space at all times, our algorithm generates a stream by traversing all tree nodes. From this stream, we continuously select nodes for pairwise comparisons, gradually converging to the optimal region. This approach minimizes memory usage while maintaining high accuracy in identifying the best arm.
>
> In terms of theoretical analysis, we acknowledge that the techniques employed are relatively standard. However, the advantage of our approach lies in its ability to provide a clear and rigorous framework that supports the algorithm's effectiveness and efficiency.
>
>
> **Q2: Line 136: Should be norm rather than absolute value, right?**
>
> Thank you for pointing this out. We will make the correction to avoid any misunderstanding by the readers.
>
> **Q3: Line 156: Definition of the mesh seems like it contains ALL elements of a cover set. Should be only one per set?**
>
> A subset $S \subset X$ is called an $\epsilon$-mesh if every point $x \in X$ is within a distance $\epsilon$ from some point $y \in S$. This ensures that each point in the space is sufficiently covered by the mesh.
>
>
> **Q4: Line 196: The use of $O$ notation for the fraction is nonstandard and confusing.**
>
> Thank you for bringing this to our attention. It is indeed inappropriate to use $O$ notation in this context. The original sentence, “Each cross exploration phase will only explore $O\left(\frac{1}{\log \log T}\right)$ of them,” should be revised to “Each cross exploration phase will only explore approximately $\frac{1}{\log \log T}$ of them.”
>
>
> **Q5: I did not find the limitations presented as sufficient and I would like to see more discussion on the downsides of the presented algorithm and future directions of research.**
>
> Thank you for highlighting this area for improvement. We will add a more comprehensive discussion about the limitations of our algorithm and potential future research directions. This will include a deeper analysis of scenarios where the algorithm may face challenges and opportunities for further enhancement and application.
>
>
>
> **Please kindly let us know if you have any concerns you find not fully addressed. We are more than happy to have a further discussion regarding them.**

---

> > ### Comment · Reviewer_mKhe · 2024-08-09
> >
> > Thank you for the detailed response. I still don't find the insights enough to increase my score, I will keep it as-is for now and wait for further discussion with the other reviewers.
> >
> > I still don't think Line 156 is correct. I understand the definition, but the notation is wrong - from the current notation it seems you take all $x_i\in X_i$, not a single element.

---

> > > ### Author Response · Authors · 2024-08-10
> > >
> > > Thank you very much for your feedback. We will amend the notation to clarify this point explicitly in the revised manuscript. We greatly appreciate your attention to detail, and we are committed to improving the precision and clarity of the definition based on your valuable feedback.

---

### Official Review · Reviewer_q7Fp · 2024-07-17

**Soundness:** 3
**Presentation:** 3
**Contribution:** 3
**Rating:** 4
**Confidence:** 4

**Summary:**

This paper studies the Lipschitz bandit problem with a memory constraint. There are two algorithms proposed by the authors. The Memory Bounded Uniform Discretization (MBUD) algorithm uses a fixed discretization over the metric space and implements a strategy which explores first and then commits to an exploitation phase. The second algorithm, called Memory Bounded Adaptive Discretization (MBAD) , swaps arms in and out of the memory while creating a mesh over the metric space adaptively (ala zooming). The authors prove upper bounds which match lower bounds from previous work for Lipschitz bandits without memory constraints while maintaining linear time complexity and constant space complexity. Finally, the authors perform experimental validation of the theoretical results on small 1-dimensional datasets.

**Strengths:**

1. Novel problem formulation in the Lipschitz bandit setting.
2. The authors show upper bounds matching with lower bounds from prior work while maintaining a memory budget on arms.
3. The concepts introduced in the paper are well explained for the most part. I did have a little trouble reading the parts about the crosscut and generating cubes but it might just be me not being familiar with prior work in the area.

**Weaknesses:**

I am unclear about the novelty and contributions of the paper. The problem formulation (limited memory) is new in the Lipschitz bandit setting but it has been studied in several papers in bandits with finite arms (as the authors point out in the related works). Moreover, the proof techniques used in the paper appear standard - MBAD is based on zooming introduced by kleinberg et al., the clean event analysis is from the recent textbook of Slivkins (and their papers), MBUD is based on an explore first strategy resembling the naive Explore-then-Commit algorithm (which trivially satisfies the O(1) memory constraint). In all, I’m not sure what specific parts of the paper are being claimed as novel vs that from prior work.

The experiments in this paper are very limited - only a 1 dimensional interval with an L1 metric. To show real world applicability, it would be nice to have results in higher dimensions and also on real world datasets (since that was the original motivation).

Minor/Typo:
I think the caption for Fig 1 should clarify what is on the X and Y axes. It is obvious from context but it would be nice to have from a readability perspective.

**Questions:**

1. From what I recall, the Kleinberg survey paper proves bounds for all metric spaces, whereas in this paper only the L1 metric on [0, 1]^d is considered. Is there any reason why similar techniques from that paper would not translate to the bounded memory setting for all metric spaces?
2. On line 229, the authors state “computational workload of the MBUD algorithm is characterized by a constant per-round operation,” but don’t the subroutines (like CROSSCUBE, GENERATECUBE etc) have time complexity depending on T? Is this statement meant to ignore sublunar factors?
3. Do we need to know the zooming dimension beforehand for MBAD? If so, it should be mentioned in the paper as it it was not clear to me.

**Limitations:**

The Lipschitz constant needs to be known beforehand to apply these algorithms.

---

> ### Author Rebuttal · Authors · 2024-08-07
>
> We sincerely appreciate your valuable time and efforts in reviewing this paper. We thank the thoughtful feedback you provided, which significantly improved the quality of this paper. For the potential concerns you bring up, we would like to answer/address them here.
>
> **Q1: I am unclear about the novelty and contributions of the paper. The problem formulation (limited memory) is new in the Lipschitz bandit setting but it has been studied in several papers in bandits with finite arms (as the authors point out in the related works) ...**
>
> We thank you for your comment. The novelty of our paper lies primarily in the design of the algorithm and the novel theoretical results we obtained. Regarding the algorithm design, our main contribution can be divided into two key insights.
>
> The first insight involves the use of metric embedding, which maps elements from one metric space to another while preserving distance relationships as much as possible. The critical aspect of metric embedding is finding a mapping that maintains the original space's distance metrics, making data processing, analysis, and understanding more efficient and intuitive. Our algorithm essentially maps the metric space to a tree structure, where each node corresponds to a cube. Traversing the nodes of this tree is equivalent to traversing the entire metric space, allowing for a more structured and efficient exploration.
>
> The second insight leverages pairwise comparisons of arms to reduce memory complexity. Rather than covering the entire space at all times, our algorithm generates a stream by traversing all tree nodes. From this stream, we continuously select nodes for pairwise comparisons, gradually converging to the optimal region. This approach minimizes memory usage while maintaining high accuracy in identifying the best arm.
>
> In terms of theoretical analysis, we acknowledge that the techniques employed are relatively standard. However, the advantage of our approach lies in its ability to provide a clear and rigorous framework that supports the algorithm's effectiveness and efficiency.
>
> **Q2: The experiments in this paper are very limited.  - only a 1 dimensional interval with an L1 metric.**
>
> We appreciate your valuable feedback. To keep the experiments straightforward and easy to understand, we initially focused on the one-dimensional case, a common practice in several related works within the field. However, we recognize the importance of demonstrating the algorithm's applicability in more complex settings. In the revised version of our paper, we will include multi-dimensional experiments to provide a more comprehensive evaluation of our algorithms.
>
> **Q3: I think the caption for Fig 1 should clarify what is on the X and Y axes.**
>
> Thank you for your suggestion. We will revise the caption to clearly specify what is represented on both the X and Y axes, enhancing the figure's clarity for readers.
>
> **Q4: Is there any reason why similar techniques from that paper would not translate to the bounded memory setting for all metric spaces?**
>
> According to Assouad’s embedding theorem, a (compact) doubling metric space can be embedded into a Euclidean space with some distortion of the metric. Therefore, considering $[0, 1]^d$ is sufficient for our paper, as it simplifies both the algorithm and the proofs. This approach is also commonly adopted in other works within the field. Our algorithm can be extended to the bounded memory setting for all (compact) metric spaces, leveraging the properties of the embedding to ensure efficient exploration and convergence in more general spaces.
>
> **Q5: On line 229, ... Is this statement meant to ignore sublunar factors?**
>
> In each round, the subroutines in the MBUD algorithm perform only a constant number of operations. Therefore, the overall time complexity remains $O(T)$.
>
> **Q6:  Do we need to know the zooming dimension beforehand for MBAD?**
>
> The algorithm requires knowledge of the covering dimension rather than the zooming dimension, which aligns more closely with practical applications. We will include a detailed explanation in the paper to clarify this aspect.
>
>
> **Please kindly let us know if you have any concerns you find not fully addressed. We are more than happy to have a further discussion regarding them.**

---

> > ### Comment · Reviewer_q7Fp · 2024-08-13
> > **Response**
> >
> > I appreciate the authors response.
> >
> > Since the paper's contribution is mainly theoretical, I'm not too worried about the state of the experiments currently (although more is always better). However, the technical significance of the paper seems limited to me from the authors responses to me and other reviewers.
> >
> > For instance, the tree based exploration of a metric space has been done in prior work in lipschitz bandits. See section 8 in [1].
> >
> > Thus, I maintain my score.
> >
> > [1] Kleinberg et al. Bandits and experts in metric spaces

---

### Comment · Area_Chair_M8ki · 2024-08-12

Dear reviewers,

Please respond to the author's response if you have not done it yet. Thank you for your attention and efforts.

Best,
AC

---

### Decision · Program_Chairs · 2024-09-25

**Decision:**

Reject

**Comment:**

This paper considers Lipschitz bandits, a version of multi-armed bandit problem where the arms are on $[0,1]^d$ and the mean reward function is $L=1$ Lipchitz on that. The authors proposed MBAD algorithm that improves time and space complexity over existing algorithms.

The problem setting is known, and the contribution would be great if it were presented appropriately. For us, the largest concern is its solidness. Rev Bcg5 considers the mathematical presentation unclear and the claimed results hard to verify. I hope the authors revise the paper based on the reviews.